# Incomplete HPV Vaccination among Individuals Aged 27–45 Years in the United States: A Mixed-Effect Analysis of Individual and Contextual Factors

**DOI:** 10.3390/vaccines11040820

**Published:** 2023-04-10

**Authors:** Victor Adekanmbi, Fangjian Guo, Christine D. Hsu, Yong Shan, Yong-Fang Kuo, Abbey B. Berenson

**Affiliations:** 1Center for Interdisciplinary Research in Women’s Health, School of Medicine, The University of Texas Medical Branch, Galveston, TX 77555, USA; 2Department of Obstetrics and Gynecology, The University of Texas Medical Branch, 301 University Blvd., Galveston, TX 77555, USA; 3Department of Biostatistics and Data Science, The University of Texas Medical Branch, Galveston, TX 77555, USA

**Keywords:** human papillomavirus, incomplete vaccination, individual factors, contextual factors, multilevel, United States

## Abstract

**Background:** In the United States, the human papillomavirus (HPV) vaccine is approved for use in individuals up to age 45. Individuals 15 years and older require three doses of the vaccine to complete the recommended dosing series. Incomplete HPV vaccination rates (i.e., one or two doses) among those over age 26, however, remain high. This study examined the independent effects of individual- and neighborhood-level factors on incomplete HPV vaccination rates in the United States (U.S.) among those aged 27–45 years. **Methods:** This retrospective cohort study used administrative data from Optum’s de-identified Clinformatics^®^ Data Mart Database to identify individuals aged 27–45 years who received one or more doses of HPV vaccine between July 2019 and June 2022. Multilevel multivariable logistic regression models were applied to the data on 7662 individuals identified as being fully or partially vaccinated against HPV, nested within 3839 neighborhoods across the U.S. **Results:** Approximately half of the patients in this study (52.93%) were not completely vaccinated against HPV. After adjusting for all other covariates in the final model, being older than 30 years old decreased the odds of not completing the HPV vaccine series. Participants living in South-region neighborhoods of the U.S. had enhanced odds of not completing the vaccine series compared with those residing in Northeast-region neighborhoods (aOR 1.21; 95% CrI 1.03–1.42). There was significant clustering of incomplete HPV vaccination rates at the neighborhood level. **Conclusions:** This study revealed that individual- and neighborhood-level factors were associated with the risk of not completing the HPV vaccine series among individuals aged 27–45 years in the U.S. Interventions to improve HPV vaccination series completion rates for this age group should take into consideration both individual and contextual factors.

## 1. Background

Although the United States (U.S.) witnessed a tremendous reduction in the incidence of human papillomavirus (HPV)-associated cancers between 2006 and 2018 after the introduction of the HPV vaccine, the burden of HPV-associated cancers remains elevated [1]. As of 2019, the incidence of cervical cancer (the most common HPV-associated cancer) in the U.S. is 6.7 per 100,000 persons [2]. Available statistics further indicate that an estimated 295,382 women in 2019 were living with cervical cancer in the U.S. [2]; moreover, about 13,000 new cases of cervical cancer are diagnosed each year, and about 4000 women die of this cancer in the U.S. annually [3].

Evidence from large observational and randomized-controlled trial (RCT) studies has shown that the available HPV vaccines reduced cervical cancer incidence by 90% in fully vaccinated girls and young adults [4,5,6,7]. However, the proportion of eligible individuals (male and female) who are up-to-date (received two doses (first dose at 0 months and second dose at 6–12 months) for those aged 9–14 years and three doses (first dose at 0 months, second dose at 1–2 months, and third dose at 6 months) for those aged 15 years through 26 years was only 13.8% in 2013 and 21.5% in 2018 [8]. As of June 2019, adults aged 27–45 years [9] have been added to the list of individuals eligible for catch-up vaccination and can receive the recommended three doses of the HPV vaccine (first dose at 0 months, second dose at 1–2 months, and third dose at 6 months) of the HPV vaccine. The decision to be vaccinated against HPV in these individuals is, however, based on shared decision-making between the healthcare providers and the patients [10].

Due to unsatisfactory rates of HPV vaccination in the U.S., the National Vaccine Advisory Committee (NVAC) was charged by the U.S. Department of Health and Human Services (HHS) to examine the factors responsible for low HPV vaccination uptake and provide recommendations on how to bolster the effectiveness of nation-wide efforts to improve HPV vaccination uptake and coverage rates [11].

Several studies [12,13,14,15,16,17,18,19] have been conducted in the U.S. on factors associated with the HPV vaccine initiation and completion of the vaccine series. Most have focused their attention on individual-level factors alone, neglecting the importance of contextual- or area-level factors. Currently, a great deal of attention is now being directed to socio-environmental factors that determine health and health-related outcomes. This has the advantage of increasing our understanding and knowledge of the social and environmental determinants of health beyond individual-level characteristics [20,21].

Few studies have examined the effects of contextual- or area-level factors on individuals who initiated and completed the HPV vaccination series, leaving out individuals who started their HPV vaccine series but did not complete it [18,19]. The exploration of individual and contextual factors associated with a health outcome together is advantageous as it prevents methodological and practical difficulties seen with examining either individual or contextual factors alone. Studies that focus on individual-level factors alone suffer from individualistic fallacy because of interwoven relationships between an individual and contextual- or area-level factors. In the same vein, studies conducted using contextual- or area-level factors alone risk having fallacy of division (ecological fallacy) because inferences from a group that an individual belongs to fail to take into consideration the unique experiences of individuals as they differ from other members of the group. In addition, previous studies have focused on adolescents and teens, leaving out adults who are older and might be more sexually active, hence, are at increased risk for new HPV infection.

The overarching aim of this research work was to determine whether individual- and neighborhood-level factors were predictors of incomplete HPV vaccination among individuals aged 27–45 in the U.S. in a single analytical framework with a view to providing up-to-date, accurate, and credible data to inform the design of programs and interventions to address the high rate of incomplete HPV vaccine series.

## 2. Methods

This study followed the Strengthening the Reporting of Observational Studies in Epidemiology (STROBE) guidelines (S1 STROBE checklist).

### 2.1. Participants and Data

We conducted this retrospective cohort study using health insurance claims from Optum’s de-identified Clinformatics^®^ Data Mart Database, which is mainly used for research. The database included data on insurance claims for vaccinations, medical services rendered, and prescriptions by United Health Group, Inc., a private health insurance company in the United States. The participants were mostly adult professional employees, with a slightly higher representation of southern states. The data on claims for medical services were obtained from medical tables, while demographic and enrollment information was obtained from member tables. Enrollment data were used to determine the continuity of medical coverage and to confirm an enrollee’s healthcare. Our study used secondary data with anonymized information that fulfilled all federal regulations and guidelines, which are clearly set out under the 45 Code of Federal Regulations (CFR) 46 regarding its use for research activities involving human subjects.

We obtained 158,636 records for participants who had a claim for HPV vaccines identified by the Current Procedural Terminology (CPT) codes 90649, 90650, and 90651. All study participants were continuously enrolled for at least 12 months before the start of the vaccine series and at least 12 months after. Twelve months of enrollment after the first HPV vaccine dose was chosen to ensure a long enough time for an enrollee to complete the vaccine series. Individuals with no HPV vaccination history were excluded from the analysis. We restricted our sample to individuals aged 27 to 45 years old who received at least one dose of the HPV vaccine. After applying all the exclusion criteria above, we selected participants who had complete sex and age data, and this resulted in 7662 participants in the study sample (see Figure 1 for participant inclusion procedure).

### 2.2. Ethical Considerations

This study analyzed de-identified datasets collected between 2019 and 2022 by Optum’s de-identified Clinformatics^®^ Data Mart Database. The conduct of this study was exempt from review by the University of Texas Medical Branch Institutional Review Board (IRB) because it was limited to the use of previously collected anonymized data.

### 2.3. Independent Variables

#### Individual-Level Factors

In this study, we considered the following individual-level factors: sex, age, smoking status, and obesity. Sex was categorized as female and male. The age of participants was categorized as 27–30, 31–40, and 41–45. For smoking, we used 1 year before and 1 year after claims data and at least 1 inpatient claim or 2 other non-drug claims of any service type with diagnosis (DX) codes or 1 common healthcare procedure coding system (HCPCS) code claim of any type. DX codes used included the following: ‘F17200’, ‘F17201’, ‘F17203’, ‘F17208’, ‘F17209’, ‘F17210’, ‘F17211’, ‘F17213’, ‘F17218’, ‘F17219’, ‘F17220’, ‘F17221’, ‘F17223’, ‘F17228’, ‘F17229’, ‘F17290’, ‘F17291’, ‘F17293’, ‘F17298’, ‘F17299’, ‘O99330’, ‘O99331’, ‘O99332’, ‘O99333’, ‘O99334’, ‘O99335’, ‘T65211A’, ‘T65212A’, ‘T65213A’, ‘T65214A’, ‘T65221A’, ‘T65222A’, ‘T65223A’, ‘T65224A’, ‘T65291A’, ‘T65292A’, ‘T65293A’, ‘T65294A’, and ‘Z720’. HCPCS codes: ‘99406’, ‘99407’, ‘G9276’, and ‘G9458’. The Charlson comorbidity index was categorized as none, one, and two or more. Smoking status was coded as non-smoker and smoker. For obesity, we used the previous year’s claims data. The DX code of E66 was used to identify obesity.

### 2.4. Contextual-Level Factors

The percentage of high school graduates and median household income in the patient’s primary ZIP code was obtained from the U.S. Census Bureau database American Community Survey (ACS) 5-year estimates from 2020. This was a record linked to health insurance claims data from Optum’s de-identified Clinformatics Data Mart database using ZIP code (aggregate data) as the unique identifier. The percentage of high school graduates was categorized as ≤88.2, 88.2 to 93.1, 93.1 to 95.9, and ≥95.9. Neighborhood household income was categorized as ≤USD 58,782, USD 58,782 to USD 76,288.5, USD 76,288.5 to USD 97,944, and ≥USD 97,944. We used the term “neighborhood” to describe clustering within the same geographical living environment. Neighborhoods were based on sharing a common ZIP code area.

### 2.5. Outcome Variable

The outcome of interest was incomplete HPV vaccination, defined as receiving one or two doses out of the recommended three doses of the HPV vaccine for this age group.

## 3. Statistical Analysis

### 3.1. Descriptive Analyses

Descriptive analysis for this study involved the use of numbers and percentage for all categorical variables to show the distribution of the outcome variable by the independent variables.

### 3.2. Modeling Approaches

We specified a two-level multilevel model for 7662 participants (level 1) nested within 3839 neighborhoods (level 2). We constructed four models. The first model was a null model that had no determinant variables and was specified to disintegrate the quantity of variation that existed at the neighborhood level. The second model adjusted for only individual-level variables, while the third model adjusted for neighborhood-level factors. The fourth model controlled for all the covariates simultaneously. A mixed-effect logistic regression model was utilized to test the association between the predictors and outcome variables.

### 3.3. Fixed Effects (Measures of Association)

The results of fixed effects (measures of association) were shown as adjusted odds ratios (aORs) with their 95% credible interval (CrI).

### 3.4. Random Effects (Measures of Variation)

The intra-cluster correlation (ICC), variance partition coefficient (VPC), and median odds ratio (MOR) were used to perform a random effects analysis. MOR is the median value of the odds ratio between a cluster with the highest likelihood of an outcome and another with the lowest likelihood of an outcome; it measures the cluster variance as OR.

### 3.5. Model Fit and Specifications

Goodness-of-fit of the models was assessed with the Bayesian deviance information criterion (DIC), and all models were well-fitted. Multicollinearity among the different independent variables was checked with the variance inflation factor (VIF). All multivariable mixed-effects modeling was performed using multilevel windows (MLwiN) software, version 3.06 [22], calling from within Stata statistical software for Windows version 17 SE using the runmlwin command [23]. The Markov Chain Monte Carlo (MCMC) computational approach was used to fit the multilevel logistic regression models. A *p*-value of <0.05 was used to define statistical significance for all measures of association assessed.

## 4. Results

### 4.1. Descriptive Statistics

The analysis involved 7662 participants who received one or more doses of the HPV vaccine (level 1) nested within 3839 neighborhoods (level 2) across the U.S. More than half of these patients had incomplete HPV vaccination rates (see Table 1). About 64% of those aged 27–30 years did not complete their HPV vaccine series. Most of the smokers (57.4%) did not complete their HPV vaccine series. Similarly, most of the individuals with one comorbid condition (55.6%) and two or more comorbid conditions (56.8%) did not complete their HPV vaccine series. Moreover, a preponderance of residents in neighborhoods with a percentage of high school graduates that were less than or equal to 88% received only one or two doses of their HPV vaccine series. More than half of those living in neighborhoods with a median household income of less than or equal to USD 58,782 (54.9%) did not complete the HPV vaccine series. Similarly, more than half who lived in neighborhoods in the southern U.S. were not fully vaccinated against HPV.

### 4.2. Multilevel Analysis of the Factors Associated with Incomplete HPV Vaccination

Table 2 shows the results of multilevel models for both individual- and neighborhood-level factors. With all factors controlled for in Model 4, participants who were 41–45 years and 31–40 years old were 54% and 43% less likely not to complete the entire series compared with those who were 27–30 years old. Participants’ sex was not statistically significantly associated with the odds of incomplete HPV vaccination. After adjusting for all other covariates, the odds of incomplete HPV vaccination increased by 25% (aOR 1.25; 95% (CrI) 0.97–1.61) in smokers when compared with non-smokers. However, this association was not statistically significant. Similarly, the odds of incomplete HPV vaccination increased by 21% in participants with one comorbid condition (aOR 1.21; 95% (CrI) 0.97–1.52) and by 18% in participants with two or more comorbid conditions (aOR 1.18; 95% (CrI) 0.90–1.54) compared with no comorbid condition, but the associations were not statistically significant. When other factors were controlled for in Model 4, context-level region of residence remained statistically significantly associated with incomplete HPV vaccination. Participants who resided in the South region were 21% (aOR 1.21; 95% CrI 1.03–1.42) more likely not to complete their HPV vaccination when compared with those in the Northeast region.

### 4.3. Random Effects Measures

There were statistically significant differences in the odds of not completing the HPV vaccine series (τ = 0.050 (0.233), *p* < 0.0001) across neighborhoods from the null model (see Table 2). The ICC indicated by the estimated intercept component variance revealed that 14.9% of the variance in the odds of incomplete HPV vaccination could be attributed to neighborhood-level factors. The variations in incomplete HPV vaccination across neighborhoods remained statistically significant, even after simultaneously controlling for both the individual- and neighborhood-level factors in the final model. The percentage change in variance in the final model indicates that the final model accounts for about 19.6% in the odds of incomplete HPV vaccination across the neighborhoods.

Results of the MOR also confirmed evidence of neighborhood contextual phenomenon modifying odds of incomplete HPV vaccination. The MOR for incomplete HPV vaccination was 1.24 in the null model; this relatively moderate MOR suggests that the clustering effect was moderate. The neighborhood-level heterogeneity in odds of incomplete HPV vaccination decreased to a MOR of 1.21 when both individual- and neighborhood-level factors were controlled for in the final model. Thus, there are moderate variations between neighborhoods in the predisposition for not completing the HPV vaccine series.

## 5. Discussion

This research work examined individual- and neighborhood-level factors as determinants of incomplete HPV vaccine series in the population studied using claims data by one of the largest private insurance companies in the U.S. This study confirms the importance of neighborhood-level variations with respect to incomplete HPV vaccine series among adults 27–45 years of age who initiated the vaccine in the U.S. In particular, the study showed that the context and region of the neighborhood in which people reside are linked to incomplete HPV vaccine series even after taking into consideration the individual-level sociodemographic risk factors.

Findings from this study show that older participants were less likely not to complete their vaccination series compared to younger participants, which is contrary to findings of previous studies [24,25,26] carried out in adolescents and young adults. A reasonable explanation could be that most younger individuals lack full insurance coverage, which may be a major barrier to HPV vaccination among young adults due to the high cost of the vaccine series. Young adults, especially those aged 26–34 years, typically have low rates of being fully insured in the U.S. as they are no longer able to be kept on their parents’ insurance as dependents [27]. However, this is very different from our study cohorts, who have full private insurance coverage. Another plausible reason could be that older participants may have more comorbid conditions that make them seek healthcare more frequently. Increased healthcare visits may result in more opportunities for vaccination and higher overall HPV vaccination completion rates. This was, however, not the case in this present study when the comorbidity index was adjusted for in our analysis. There was no statistically significant association between having one or more comorbid conditions and not completing the HPV vaccine series when compared with having no comorbid condition. An alternative explanation could be because of low health-seeking behaviors commonly seen in younger individuals. Most young adults will only seek healthcare when they are seriously ill. This finding is consistent with the results of previous studies [28,29] that have compared healthcare-seeking behaviors between the elderly and young adults.

The effects of neighborhood-level factors on incomplete HPV vaccination status are highlighted in this study. Our findings indicate that the likelihood of not being fully vaccinated against HPV is higher for residents in neighborhoods in the South region compared with those living in neighborhoods in the Northeast. This could be due to the high rate of vaccine hesitancy among the people living in the South [30,31]. Rurality is another potential reason that could explain low HPV vaccine series completion in the South. Most southern states have been noted to consist of a large proportion of rural counties compared with states in other U.S. regions [32,33,34]. Some of the reasons that were identified as likely associated with low vaccination uptake and completion among rural dwellers are lower income and less knowledge and awareness of HPV vaccination and its benefits [32,33,34]. There is, therefore, a need to design policies and interventions to improve HPV vaccine uptake and completion in the South to lower the burden of HPV-associated diseases and cancers.

Approximately half of the participants in this study did not complete their vaccination series, which is consistent with the findings of previous studies [35,36,37] on HPV vaccination and other vaccine types. A lack of needed information was consistently reported among study participants who did not complete the vaccination series from most of these studies. Prior studies conducted in the U.S. and Italy [35,36] identified the lack of information about the benefits of HPV vaccines as one of the key drivers of not completing the vaccination series. In a similar study carried out in The Netherlands, a lack of adequate information was identified as the predictor of HPV vaccine uptake and completion [37]. The provision of clear information about the benefits of HPV vaccination and the risks of developing HPV infection and associated diseases are vital communication strategies to improve the uptake and completion of the HPV vaccination series [38].

Previous studies [35,36] have also reported fear of adverse events from the HPV vaccine as one of the reasons why people refuse HPV vaccination. Patients living with comorbidities in these studies were particularly critical of the roles of pharmaceutical companies in HPV vaccination promotion. More importantly, participants were afraid of developing long-term side effects that may worsen their conditions [35,37].

Another reason people may refuse to initiate and complete their HPV vaccination series could be due to logistical reasons. Participants of a cross-sectional study that examined this topic stated that the distance to the vaccination centers was too far for them and that they did not have time due to the pressure of work to go to the vaccination centers to initiate as well as complete their vaccine series [39]. A better understanding and knowledge of the benefits of HPV vaccination on the parts of patients may enhance acceptability, initiation, and completion of the vaccine series in those reporting logistical challenges and lack of time as the barriers to initiation and completion of their vaccine series. Moreover, improved accessibility to vaccination services, such as offering vaccination services over the weekends, could also increase vaccination initiation and completion rates.

Future studies should be conducted to examine the effects of the numerous unexplored/under-explored cluster-level risk factors associated with incomplete HPV vaccination, which this current study could not investigate.

### Limitations and Strengths

Findings from this study should be interpreted in the context of its limitations. One limitation of this study, as with most retrospective cohort studies, is the use of existing data, which provided us with limited data for us to adjust for more risk factors, such as race/ethnicity, marital status, and socioeconomic or other demographic factors. This may result in residual confounding. Second, information on our covariate of smoking and obesity came from diagnosis codes and relied on insurance claims for outpatient and hospitalization services, which are not always 100% accurate or complete. Third, we were unable to determine whether our study participants may have been partially or fully vaccinated through the second insurance provider. Fourth, we did not adjust for rural/urban residency, preventive behaviors, and healthcare services usage. Lastly, Optum’s de-identified Clinformatics^®^ Data Mart Database included only privately insured participants, and these results are not generalizable to publicly insured individuals.

Despite the limitations of this study, it has several strengths that are worth mentioning. One important strength of this current study is the use of a large national database of privately insured enrollees. The database contains longitudinal tracking at the individual patient level, which provides at least one year of follow-up data. This will make the findings of the study to be relevant to the study population and generalizable if applied to similar populations. In addition, being a population-based study with a large sample size is an important strength of this study. Furthermore, no studies to our knowledge have looked at factors associated with HPV vaccine series completion in the catch-up age range of 27–45 years old. Lastly, the use of a mixed-effects analysis for this study revealed other contextual-level factors aside from the individual-level factors that were responsible for the variations in incomplete HPV vaccination, which a standard single-level regression model would not be able to reveal.

## 6. Conclusions

This study revealed that individual- and neighborhood-level factors are significantly associated with failure to complete the HPV vaccine series among adults who previously received one or two doses. Both individual- and contextual-level characteristics should be taken into consideration during the design, planning, and implementation of policies, programs, and interventions aimed at improving the completion of the HPV vaccine series among adults in the U.S.

## Figures and Tables

**Figure 1 vaccines-11-00820-f001:**
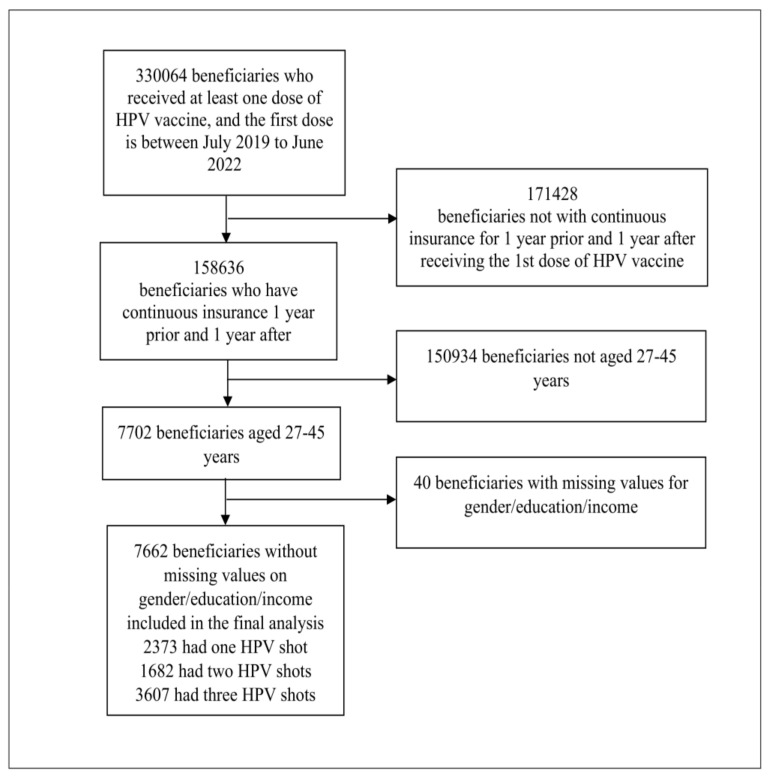
Optum data; HPV, Human Papillomavirus.

**Table 1 vaccines-11-00820-t001:** HPV vaccination status at different levels of independent variables.

Variables	Incomplete (N, %)	Complete (N, %)	Total (N, %)	*p*-Value
**Individual-level factors**	**4055 (52.92)**	**3607 (47.08)**	**7662 (100)**	
**Sex**				
Male	1425 (54.85)	1173 (45.15)	2598 (100)	
Female	2630 (51.94)	2434 (48.06)	5064 (100)	<0.05
**Age in years**				
27–30	1174 (64.33)	651 (35.67)	1825 (100)	
31–40	2144 (50.88)	2070 (49.12)	4214 (100)	
41–45	737 (45.41)	886 (54.59)	1623 (100)	<0.0001
**Obesity**				
Normal	3910 (52.94)	3476 (47.06)	7386 (100)	
Obese	145 (52.54)	131 (47.46)	276 (100)	0.896
**Smoking**				
Non-smoker	3900 (52.76)	3492 (47.24)	7392 (100)	
Smoker	155 (57.41)	115 (42.59)	270 (100)	0.133
**Charlson comorbidity index**				
None	3729 (52.65)	3353 (47.35)	7082 (100)	
One	189 (55.92)	149 (44.08)	338 (100)	
Two or more	137 (56.61)	105 (43.39)	242 (100)	0.254
**Contextual-level factors**	**N = 3839**			
**Percentage of high school graduate**				
≤88.2	1048 (54.53)	874 (45.47)	1922 (100)	
88.2–93.1	1039 (53.17)	915 (46.83)	1954 (100)	
93.1–95.9	1020 (53.12)	900 (46.88)	1920 (100)	
≥95.9	948 (50.80)	918 (49.20)	1866 (100)	0.143
**Household income**				
≤USD 58,782	1051 (54.85)	865 (45.15)	1916 (100)	
USD 58,782–76,288.5	1029 (53.73)	886 (46.27)	1915 (100)	
USD 76,288.5–97,944	1008 (52.61)	908 (47.39)	1916 (100)	
≥USD 97,944	967 (50.50)	948 (49.50)	1915 (100)	<0.05
**Region**				
Northeast	489 (50.62)	477 (49.38)	966 (100)	
Mid-west	994 (52.96)	883 (47.04)	1877 (100)	
West	1518 (52.00)	1401 (48.00)	2919 (100)	
South	1054 (55.47)	846 (44.53)	1900 (100)	<0.05

N—number of participants. %—proportion (percentage).

**Table 2 vaccines-11-00820-t002:** Factors associated with incomplete HPV vaccination identified by multilevel multivariable logistic regression models.

Variable	Model 1 ^a^	Model 2 ^b^	Model 3 ^c^	Model 4 ^d^
Fixed Effects	OR (CrI)	aOR (CrI)	aOR (CrI)	aOR (CrI)
**Individual-level factors**				
**Sex**				
Female (vs. Male)		1.09 (0.99–1.20)		1.09 (0.99–1.20)
**Age in years**				
27–30		1 (reference)		1 (reference)
31–40		0.57 (0.51–0.64)		0.57 (0.51–0.64)
41–45		0.45 (0.39–0.52)		0.46 (0.40–0.53)
**Obesity**				
No obese		1 (reference)		1 (reference)
Obese		1.01 (0.79–1.30)		1.01 (0.79–1.30)
**Smoking status**				
Non-smoker		1 (reference)		1 (reference)
Smoker		1.24 (0.96–1.59)		1.22 (0.95–1.57)
**Charlson comorbidity index**				
None		1 (reference)		1 (reference)
One		1.21 (0.97–1.52)		1.21 (0.97–1.52)
Two or more		1.20 (0.92–1.57)		1.18 (0.90–1.54)
**Contextual-level factors**				
**Percentage of high school graduate**				
≤88.2			1 (reference)	1 (reference)
88.2–93.1			0.97 (0.85–1.11)	0.99 (0.86–1.13)
93.1–95.9			1.01 (0.87–1.17)	1.01 (0.88–1.18)
≥95.9			0.93 (0.79–1.09)	0.96 (0.81–1.13)
**Household income**				
≤USD 58,782			1 (reference)	1 (reference)
USD 58,782–76,288.5			0.95 (0.83–1.09)	0.97 (0.84–1.11)
USD 76,288.5–97,944			0.91 (0.79–1.06)	0.95 (0.82–1.11)
≥USD 97,944			0.85 (0.72–1.00)	0.90 (0.76–1.07)
**Region**				
Northeast			1 (reference)	1 (reference)
Mid-west			1.08 (0.91–1.27)	1.07 (0.90–1.26)
West			1.03 (0.88–1.20)	1.03 (0.88–1.22)
South			1.22 (1.03–1.43)	1.21 (1.03–1.40)
**Measures of variation**				
Variance (SE)	0.050 (0.223)	0.048 (0.218)	0.042 (0.204)	0.040 (0.199)
Explained variation (%)	Reference	3.7	16.3	19.6
Intra-cluster correlation (%)	14.9	14.4	12.5	12.0
MOR	1.24	1.23	1.21	1.21
**Model fit statistics**				
Bayesian DIC	10,611.5	10,522.0	10,673.7	10,590.7

^a^ Model 1 is the null model, a baseline model without any predictor variables. ^b^ Model 2 is the individual factors. ^c^ Model 3 is for contextual-level factors. ^d^ Model 4 is for the final model. Abbreviations: CrI, credible interval; aOR, adjusted odds ratio; SE, standard error; MOR, median odds ratio; and DIC, deviance information criterion.

## Data Availability

The data used in this study is available in the Optum’s Clinformatics^®^ Data Mart Database. Optum’s Clinformatics^®^ Data Mart has established an application process to be followed by anyone who would like to access data via https://cdn-aem.optum.com/content/dam/optum4/resources/pdf/clinformatics-data-mart.pdf Mart.

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
