# Peer review of "Incomplete HPV Vaccination among Individuals Aged 27–45 Years in the United States: A Mixed-Effect Analysis of Individual and Contextual Factors"

_vaccines, 2023, doi:10.3390/vaccines11040820_

Round 1

Author Response

In my opinion, this topic analyzed is interesting enough to attract readers’ attention. The goal of this retrospective cohort study is the exam of the independent effects of individual and neighborhood level factors on incomplete HPV vaccination rates in the United States among women aged 27-45 years. I think that the abstract of this article is very clear and well structured.

Reply: Thanks for your kind remarks and constructive comments.

In my opinion, the discussion could be studied in deep and extended. Maybe, it could be useful the evaluation of the state of the art of preventive strategies and protocols for gynecological cancers. I suggest these articles to get deeper in the topic: PMID: 35742340, PMID: 35455328, Because of these reasons, the article should be revised and completed.

Reply: Thanks for this useful suggestion. We have now expanded our discussion section as advised.

Tables are clear and interesting. Considered all this points, I think it could be of interest for the readers and, in my opinion, it deserves the priority to be published after minor revisions.

Reply: Thank you for your kind words about our manuscript.

Reviewer 2 Report

Incomplete vaccinations remain high in adults.Independent effects of individual and neighborhood level factors are searched.The study revealed that individual and neighborhood level factors associated with failure to complete HPV vaccine series.Wery well conducted study...

1- the importance of neighborhood level variations with respect to incomplete HPV vaccine series among adults 27-45 years of age who initiated the vaccine  in the US. 2-relevant and it has scientific merit fiiling the gap of the subject. 3-it looked at factors associated  with HPV vaccine series completion in the catch up age range of 27–45 years old. 4-I strongly advice to conduct the same research in the same study group for the next 5 years. 5-The discussion is greatly appreaciated. 6-References are adequate.

Author Response

Incomplete vaccinations remain high in adults. Independent effects of individual and neighborhood level factors are searched. The study revealed that individual and neighborhood level factors associated with failure to complete HPV vaccine series. Very well conducted study...

1- the importance of neighborhood level variations with respect to incomplete HPV vaccine series among adults 27-45 years of age who initiated the vaccine in the US. 2-relevant and it has scientific merit filling the gap of the subject. 3-it looked at factors associated with HPV vaccine series completion in the catch-up age range of 27–45 years old. 4-I strongly advice to conduct the same research in the same study group for the next 5 years. 5-The discussion is greatly appreciated. 6-References are adequate

Reply: We thank this reviewer for their encouraging comments about our manuscript. We have also thought of conducting long term longitudinal study on this topic when more data become available.

Reviewer 3 Report

The paper analyzes the factors associated with incomplete HPV vaccination among US adults aged 27-45, including both individual factors and contextual factors. I have a few comments concerning the analysis.

The incomplete vaccination rate in the sample is very high (>50%). I find the proportion quite surprising. Why would such a large number of people get the first dose and then refuse to get the others? It would be good if you could contextualize a bit more this finding. In particular:

-        Is this a specific feature of the sample (employed with private insurance) or is it representative of the overall population? Can you compare it to statistics from other sources about HPV incomplete vaccination rates? Or perhaps compare it to the incompletion rate of other vaccines?

-        Is there any literature or anecdotal evidence about the possible reasons why people may not complete their HPV vaccine series? Perhaps people are experiencing side effects after the first dose, or they report not having the time to get vaccinated for a second dose.

-        It would also be useful to discuss the (non-HPV) literature on vaccine hesitancy and partial vaccination. What are the main drivers of partial vaccination in the context of other vaccines and how do they compare to HPV vaccines? Is there any policy recommendation that can be derived from that literature and that can be applied to HPV vaccines?

The paper misses some key details about HPV vaccines that may not be common knowledge, e.g., the recommended schedule of HPV vaccine doses, the relative effectiveness of different doses, the price of HPV vaccine and whether it usually is covered by insurance, the setting where HPV vaccine can be received (e.g., pharmacy, hospital).

The sample is restricted to individuals who are enrolled for at least 12 months after vaccine series initiation. Did you try to extend the time horizon beyond 12 months? I am wondering whether people are refusing to get vaccinated or postponing the vaccination. Did you check the timing of vaccination among those who complete the series? I mean, do they follow the recommended schedule on average or are there many people who delay the administration of the second or third dose? If there are many people who wait months before getting the second or third dose, then maybe the 12 months window is too short.

In the discussion section, the authors mention that the lower vaccination rates in the South region may be due to larger levels of rurality. If rural residence is deemed to be a strong predictor of vaccination behavior, why didn’t you control for that directly in the regression? Given knowledge of the zip code, it should be easy to determine the rurality of the location.

The section on data description is a bit misleading (lines 92-101). At first reading, it sounded as if data on education and income were at the individual level and provided in the Optum database. I would suggest to more clearly state that those data come from ACS.

I am not sure how to interpret the results and discussion about the contextual factors. It’s possible that those factors are significant only because you do not have individual-level data on income and education. Thus, it’s unclear whether you are capturing a neighborhood effect that is distinct from the individual effect, or whether neighborhood characteristics are simply a proxy for missing individual characteristics. I mean, individuals who live in rich zip codes are also very likely to be rich, and, if income is positively correlated with vaccination, they are very likely to complete the series. If that’s the case, being rich is correlated with vaccination, while living in a rich neighborhood may have no impact per se. Could you explain a bit better your claims about the importance of context? Are you saying that context shapes people’s beliefs and norms, and thus vaccination behavior? Or that location of healthcare facilities is neighborhood-dependent? I am not sure you can make either claims with the data at hand.

The regression controls for some variables related to health status (obesity and smoking). What was the rationale for choosing these specific variables? Did you try controlling for other health-related variables or constructing an overall health index? You could also control for preventive behavior, e.g., whether those individuals took the flu vaccine. My hypothesis is that individuals who show preventive behavior in other dimensions are also more likely to complete the HPV vaccination series.

In the discussion, the authors mention that frequency of interactions with the healthcare systems may explain vaccination uptake. Can you control for that in the regression? For example, by measuring the usage of healthcare services.

Author Response

The paper analyzes the factors associated with incomplete HPV vaccination among US adults aged 27-45, including both individual factors and contextual factors. I have a few comments concerning the analysis.

The incomplete vaccination rate in the sample is very high (>50%). I find the proportion quite surprising. Why would such a large number of people get the first dose and then refuse to get the others? It would be good if you could contextualize a bit more this finding. In particular:

-        Is this a specific feature of the sample (employed with private insurance) or is it representative of the overall population? Can you compare it to statistics from other sources about HPV incomplete vaccination rates? Or perhaps compare it to the incompletion rate of other vaccines?

Reply: Thank you for pointing this out. Similar results to our study were reported by previous studies that have examined this association. The Optum data that we used is representative of the population (working class adults with private health insurance) studied but not representative of the overall US population.

-        Is there any literature or anecdotal evidence about the possible reasons why people may not complete their HPV vaccine series? Perhaps people are experiencing side effects after the first dose, or they report not having the time to get vaccinated for a second dose.

Reply: Thank you for this relevant question. We have added more likely reasons for not completing HPV vaccination series reported by similar previous studies into the discussion section “see page 7-8, lines 273-299”

-        It would also be useful to discuss the (non-HPV) literature on vaccine hesitancy and partial vaccination. What are the main drivers of partial vaccination in the context of other vaccines and how do they compare to HPV vaccines? Is there any policy recommendation that can be derived from that literature and that can be applied to HPV vaccines?

      Reply: Thank you for asking this question. Based on the findings of our study, the main drivers of incomplete HPV vaccination series are young age and region of residency of the study participants. We have also added more drivers of incomplete HPV vaccination series as well as other vaccines from previous studies into the discussion section as suggested by this reviewer “see pages 7-8, lines 273-299”.

The paper misses some key details about HPV vaccines that may not be common knowledge, e.g., the recommended schedule of HPV vaccine doses, the relative effectiveness of different doses, the price of HPV vaccine and whether it usually is covered by insurance, the setting where HPV vaccine can be received (e.g., pharmacy, hospital).

Reply: Thank you for asking this question. The recommended schedule of HPV vaccine doses for different age groups in the US have been captured under the background section “see pages 1-2, lines 44-48”. Information about price of vaccine and settings where vaccine can be received were not available in the dataset used.

The sample is restricted to individuals who are enrolled for at least 12 months after vaccine series initiation. Did you try to extend the time horizon beyond 12 months? I am wondering whether people are refusing to get vaccinated or postponing the vaccination. Did you check the timing of vaccination among those who complete the series? I mean, do they follow the recommended schedule on average or are there many people who delay the administration of the second or third dose? If there are many people who wait months before getting the second or third dose, then maybe the 12 months window is too short.

Reply: Thank you for asking this question. Each participant was given minimum of 12 months after initiation of the vaccination series to complete the vaccination series. The guideline from center for disease control and prevention (CDC) recommends HPV vaccination series for catch up vaccination in the US to be completed within 6 months of initiation (1st dose at 0, 2nd dose at 1-2 and 3rd dose at 6 months).

In the discussion section, the authors mention that the lower vaccination rates in the South region may be due to larger levels of rurality. If rural residence is deemed to be a strong predictor of vaccination behavior, why didn’t you control for that directly in the regression? Given knowledge of the zip code, it should be easy to determine the rurality of the location.

Reply: Thanks for your comment. We agree with this reviewer that adjusting for rural/urban residency would have been a good idea. We have added this as one of the limitations of the study.

The section on data description is a bit misleading (lines 92-101). At first reading, it sounded as if data on education and income were at the individual level and provided in the Optum database. I would suggest to more clearly state that those data come from ACS.

Reply: Thanks for this suggestion. We have now more clearly described that the education and income variables came from ACS and not from Optum data under the variable description section of the manuscript “see page 3, lines 134-135”.

I am not sure how to interpret the results and discussion about the contextual factors. It’s possible that those factors are significant only because you do not have individual-level data on income and education. Thus, it’s unclear whether you are capturing a neighborhood effect that is distinct from the individual effect, or whether neighborhood characteristics are simply a proxy for missing individual characteristics. I mean, individuals who live in rich zip codes are also very likely to be rich, and, if income is positively correlated with vaccination, they are very likely to complete the series. If that’s the case, being rich is correlated with vaccination, while living in a rich neighborhood may have no impact per se. Could you explain a bit better your claims about the importance of context? Are you saying that context shapes people’s beliefs and norms, and thus vaccination behavior? Or that location of healthcare facilities is neighborhood-dependent? I am not sure you can make either claims with the data at hand.

Reply: The contextual factors adjusted for in this study represent median values for the neighborhoods studied and the neighborhood effects can cover everyone residing in the neighborhood represented by zip code area. In addition, context where people live shapes people’s beliefs and norms as well as vaccine behavior as you rightly pointed out. In multilevel modelling, the random effects have a way of decomposing the amount of variance existing at the cluster level, even with no determinant variable adjusted for i.e empty or null model (See Table 2). With the addition of more variables, the random effects estimates will start reducing in value. Furthermore, we adopted the concept of frailty effect to explain contextual effects affecting health outcomes from the works of Clayton1, Vaupel et al2 and Sastry3 which in this study refers to a participant’s predisposition to the risk of not completing HPV vaccination series that are inherent part of the neighborhood. According to Sastry, this model captures the total effects of all factors that influence the participant’s likelihood of having a health outcome that are not captured in the model that accounts for the measures of association (fixed effects). Because measures of association model can account for the observed predictors, the frailty effects stand for unobserved/unmeasured or unmeasurable effects.

The regression controls for some variables related to health status (obesity and smoking). What was the rationale for choosing these specific variables? Did you try controlling for other health-related variables or constructing an overall health index? You could also control for preventive behavior, e.g., whether those individuals took the flu vaccine. My hypothesis is that individuals who show preventive behavior in other dimensions are also more likely to complete the HPV vaccination series.

Reply: Thanks for your comment. We selected the health status variables based on review of existing literature4 and our clinical knowledge. We agree with this reviewer that adjusting for Charlson comorbidity index will be a good idea and based on your comment, we have now controlled for comorbidity index which was not statistically significantly associated with our outcome of interest. We have added not controlling for preventive behaviors as one of the limitations of this study.

In the discussion, the authors mention that frequency of interactions with the healthcare systems may explain vaccination uptake. Can you control for that in the regression? For example, by measuring the usage of healthcare services.

Reply: We did not control for health care services usage in our analysis because the focus of our study was on individual and contextual factors associated with not completing HPV vaccination series in those who initiated HPV vaccination series and did not complete the series. Our study did not focus on factors associated with vaccination uptake. Thank you.

References

  1. Clayton DG. A Model for Association in Bivariate Life Tables and Its Application in Epidemiological Studies of Familial Tendency in Chronic Disease Incidence. Biometrika. 1978;65(1):141-151.
  2. Vaupel JW, Manton KG, Stallard E. The Impact of Heterogeneity in Individual Frailty on the Dynamics of Mortality. Demography. 1979;16(3):439-454.
  3. Sastry N. Family-Level Clustering of Childhood Mortality Risk in Northeast Brazil. Population Studies. 1997;51(3):245-261.
  4. Harris JA, Garrett AA, Akers AY. Obesity and Disparities in Human Papillomavirus Vaccination for U.S. Adolescent Girls and Young Women. Womens Health Issues. 2019 Jan-Feb;29(1):31-37. doi: 10.1016/j.whi.2018.09.007. Epub 2018 Nov 13.